# Efficient single-copy HDR by 5′ modified long dsDNA donors

**Jose Arturo Gutierrez-Triana[†‡], Tinatini Tavhelidse[†], Thomas Thumberger[†], Isabelle Thomas, Beate Wittbrodt, Tanja Kellner, Kerim Anlas, Erika Tsingos, Joachim Wittbrodt***

Centre for Organismal Studies, Heidelberg University, Heidelberg, Germany

**Abstract** CRISPR/Cas9 efficiently induces targeted mutations via non-homologous-end-joining but for genome editing, precise, homology-directed repair (HDR) of endogenous DNA stretches is a prerequisite. To favor HDR, many approaches interfere with the repair machinery or manipulate Cas9 itself. Using Medaka we show that the modification of 5′ ends of long dsDNA donors strongly enhances HDR, favors efficient single-copy integration by retaining a monomeric donor conformation thus facilitating successful gene replacement or tagging.
DOI: https://doi.org/10.7554/eLife.39468.001

***For correspondence:**
jochen.wittbrodt@cos.uni-heidelberg.de

[†]These authors contributed equally to this work

**Present address:** [‡]Escuela de Microbiología, Facultad de Salud, Universidad Industrial, Santander, Colombia

**Competing interests:** The authors declare that no competing interests exist.

## Introduction

The implementation of the bacterial CRISPR/Cas9 system in eukaryotes has triggered a quantum leap in targeted genome editing in literally any organism with a sequenced genome or targeting region (*Cong et al., 2013*; *Jinek et al., 2012*; *Mali et al., 2013*). Site-specific double-strand breaks (DSBs) are catalyzed by the Cas9 enzyme guided by a single RNA with a short complementary region to the target site. In response, the endogenous non-homologous end joining (NHEJ) DNA repair machinery seals the DSB. Since perfect repair will restore the CRISPR/Cas9 target site, mutations introduced by imprecise NHEJ are selected for.

To acquire precise genome editing the initial DSB should be fixed via the homology-directed repair (HDR) mechanism which is preferentially active during the late S/G2 phase of the cell cycle (*Hustedt and Durocher, 2017*). Donor DNA templates with flanking regions homologous to the target locus are used to introduce specific mutations or particular DNA sequences. Injected (linear) dsDNA rapidly multimerizes (*Winkler et al., 1991*), which likely also happens in CRISPR/Cas9 based approaches. Additionally, the high activity of NHEJ re-ligating CRISPR/Cas9 mediated DSBs can multimerize injected (linear) dsDNA donor templates. This poses a problem since consequently, the precise HDR-mediated recombination of single-copy donor templates is rather rare.

Several strategies have been followed to avoid NHEJ and/or favor HDR. NHEJ was interfered with by pharmacological inhibition of DNA ligase IV (*Maruyama et al., 2015*). Conversely, HDR was meant to be favored by fusing the HDR-mediating yeast protein Rad52 to Cas9 (*Wang et al., 2017*). Similarly, removal of Cas9 in the G1/S phase (*Gutschner et al., 2016*) by linking it to the N-terminal region of the DNA replication inhibitor Geminin, should restrict the introduction of double-strand cuts to the G2 phase, when HDR is most prominently occurring (*Hustedt and Durocher, 2017*). Those approaches improved HDR-mediated integration of the homology flanks, while they did not tackle reported integration of multimers (*Auer et al., 2014*) arising after injection of dsDNA templates such as plasmids or PCR products (*Winkler et al., 1991*).

**eLife digest** CRISPR/Cas9 technology has revolutionized the ability of researchers to edit the DNA of any organism whose genome has already been sequenced. In the editing process, a section of RNA acts as a guide to match up to the location of the target DNA. The enzyme Cas9 then makes a cut in both strands of the DNA at this specific location. New segments of DNA can be introduced to the cell, incorporated into DNA 'templates'. The cell uses the template to help it to heal the double-strand break, and in doing so adds the new DNA segment into the organism's genome.

A drawback of CRISPR/Cas9 is that it often introduces multiple copies of the new DNA segment into the genome because the templates can bind to each other before being pasted into place. In addition, some parts of the new DNA segment can be missed off during the editing process. However, most applications of CRISPR/Cas9 – for example, to replace a defective gene with a working version – require exactly one whole copy of the desired DNA to be inserted into the genome.

In order to achieve more accurate CRISPR/Cas9 genome editing, Gutierrez-Triana, Tavhelidse, Thumberger et al. attached additional molecules to the end of the DNA template to shield the DNA from mistakes during editing. The modified template was used to couple a stem cell gene to a reporter that produces a green fluorescent protein into the genome of fish embryos. The fluorescent proteins made it easy to identify when the coupling was successful.

Gutierrez-Triana et al. found that the additional molecules prevented multiple templates from joining together end to end, and ensured the full DNA segment was inserted into the genome. Furthermore, the results of the experiments showed that only one copy of the template was inserted into the DNA of the fish. In the future, the new template will allow DNA to be edited in a more controlled way both in basic research and in therapeutic applications.

DOI: https://doi.org/10.7554/eLife.39468.002

## Results

To enhance HDR without interfering with the endogenous DNA repair machinery, we aimed at establishing DNA donor templates that escape multimerization or NHEJ events. We thus blocked both 5′ends of PCR amplified long dsDNA donor cassettes using 'bulky' moieties like Biotin, Amino-dT (A-dT) and carbon spacers (e. g. Spacer C3, SpC3). This should shield the DNA donor from multi-merization and integration via NHEJ, thus favoring precise and efficient single-copy integration via HDR (*Figure 1A*).

We first addressed the impact of the donor 5' modification on the formation of multimers in vivo. We injected modified and unmodified dsDNA donors into one-cell stage medaka (*Oryzias latipes*) embryos and analyzed the conformational state of the injected material during zygotic development. dsDNA donors were generated by PCR employing 5' modified and non-modified primers respectively, and by additionally providing traces of DIG-dUTP for labeling of the resulting PCR product. The conformation of the injected dsDNA donors was assayed in the extracted total DNA after 2, 4 and 6 hr post-injection, respectively. The DNA was size fractionated by gel electrophoresis and donor DNA conformation was detected after blotting the DNA to a nylon membrane by anti-DIG antibodies (*Figure 1B*). In unmodified DIG-labelled control donors, we uncovered multimerization already at 2 hr post-injection as evident by a ladder of labeled donor DNA representing different copy number multimers (*Figure 1B*). In contrast, Biotin and SpC3 modification of DIG-labelled donors prevented multimerization within six hours post-injection (*Figure 1B*). dsDNA donors established by A-dT modified primes, however, multimerized and produced similar results as unmodified DIG-labelled dsDNA donors (*Figure 1B*). Our results reveal that Biotin and SpC3 5' modifications efficiently prevent donor multimerization in vivo. While strongly blocking multimerization, the 5' modification of dsDNA did not apparently enhance the stability of the resulting dsDNA (compare modified and unmodified donors over time, *Figure 1B*).

To test whether 5' modification not only reduces the degree of multimerization but also impacts on single-copy HDR-mediated integration of long dsDNA donors, we designed *gfp* containing donor cassettes for an immediate visual readout. We generated *gfp* in-frame fusion donors for four different genes: the retinal homeobox transcription factors *rx2* and *rx1* (*Reinhardt et al., 2015*), the non-

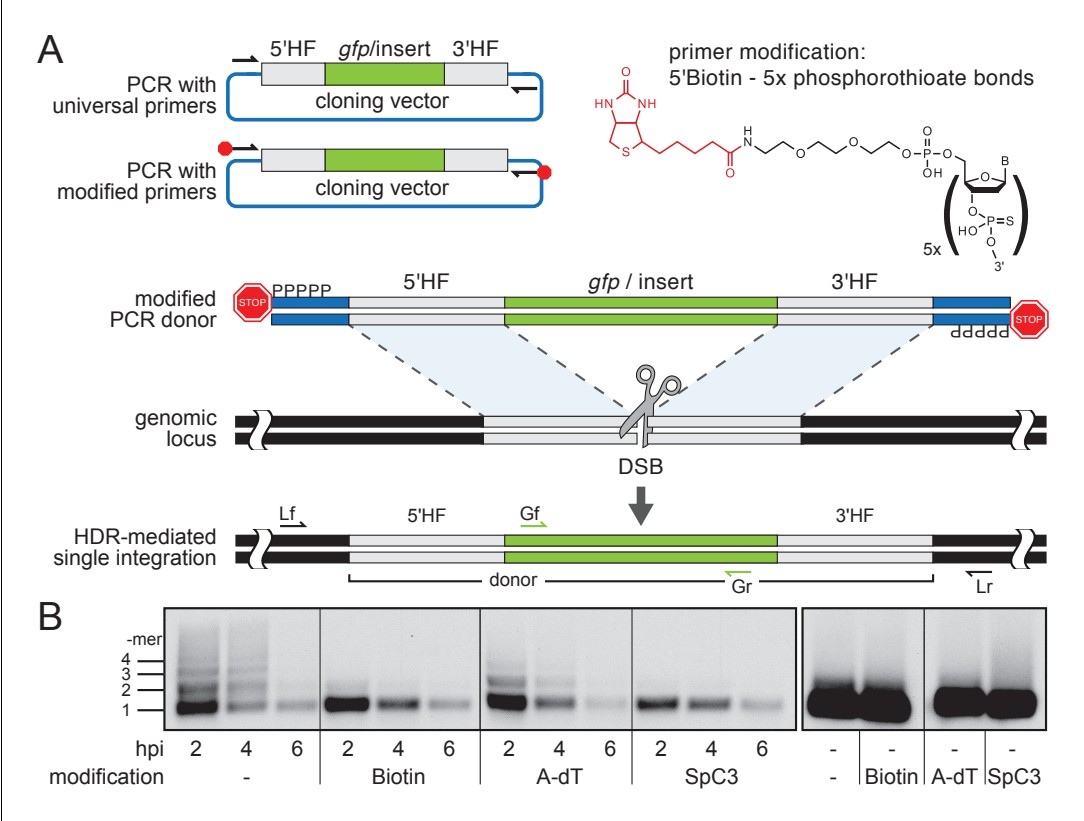

**Figure 1.** Modification of 5' ends of long dsDNA fragments prevents in vivo multimerization. (**A**) Schematic representation of long dsDNA donor cassette PCR amplification with universal primers (black arrows) complementary to the cloning vector backbone outside of the assembled donor cassette (e. g. *gfp* with homology flanks). Bulky moieties like Biotin at the 5' ends of both modified primers (red octagon) prevent multimerization/NHEJ of dsDNA, providing optimal conditions for HDR-mediated single-copy integration following CRISPR/Cas9-introduced DSB at the target locus (grey scissors). Representation of locus (Lf/Lr) and internal *gfp* (Gf/Gr) primers for PCR genotyping of putative HDR-mediated *gfp* integration events. (**B**) Southern blot analysis reveals the monomeric state of injected dsDNA fragments in vivo for 5' modification with Biotin or Spacer C3. Long dsDNAs generated with control unmodified primers or Amino-dT attached primers multimerize as indicated by a high molecular weight ladder apparent already within two hours post-injection (hpi). Note: 5' moieties did not enhance the stability of injected DNA.
DOI: https://doi.org/10.7554/eLife.39468.003

The following figure supplement is available for figure 1:

**Figure supplement 1.** Schematic representation of the donor plasmids.
DOI: https://doi.org/10.7554/eLife.39468.004

muscle cytoskeletal *beta-actin* (*actb*) (*Stemmer et al., 2015*) and the *DNA methyltransferase 1* (*dnmt1*). Donor cassettes contained the respective 5' homology flank (HF) (462 bp for *rx2*, 430 bp for *rx1*, 429 bp for *actb*, 402 bp for *dnmt1*), followed by the in-frame *gfp* coding sequence, a flexible linker in case of *rx2*, *rx1* and *dnmt1*, and the corresponding 3' HF (414 bp for *rx2*, 508 bp for *rx1*, 368 bp for *actb*, 405 bp for *dnmt1*; schematic representation in *Figure 1A*, *Figure 1—figure supplement 1* for detailed donor design). To amplify the long dsDNA donors we employed a pair of universal primers (5' modified or unmodified as control) complementary to the backbone of the cloning vectors (pDestSC-ATG [*Kirchmaier et al., 2013*] or pCS2+ [*Rupp et al., 1994*]) encompassing the entire assembled donor cassette (*Figure 1—figure supplement 1*).

Modified or unmodified long dsDNA donors were subsequently co-injected together with *Cas9* mRNA and the respective locus-specific sgRNA into medaka one-cell stage zygotes (*Loosli et al., 1999*; *Rembold et al., 2006*). For all four loci (and all 5' modifications) tested we observed efficient targeting as apparent by the GFP expression within the expected expression domain (*Figure 2A*, *Supplementary file 1*, *Figure 2—figure supplement 1*).

The survival rates of embryos injected with Biotin and SpC3 5' modified dsDNA donors did not differ significantly from embryos injected with the unmodified dsDNA control donors (*Supplementary file 1*, *Figure 2—figure supplement 1*). In contrast, the injection of the A-dT 5' modified dsDNA donors resulted in high embryonic lethality (*Supplementary file 1*, *Figure 2—figure supplement 1*).

We next analyzed the frequency of single-copy HDR events following a careful, limited cycle PCR approach on genomic DNA of GFP expressing embryos injected with unmodified, Biotin or SpC3 5' modified dsDNA donors. Our approach allowed distinguishing alleles without *gfp* integration (i.e. size of wild-type locus) from those generated by HDR and NHEJ respectively and addressed the size of the integration by a locus spanning PCR with a reduced number of PCR cycles (<=30) to omit in vitro fusion-PCR artefacts (own data and [*Won and Dawid, 2017*]). To determine the predictive power of GFP expression for perfect integration, we genotyped randomly selected, GFP-expressing embryos using locus primers (Lf/Lr) located distal to the utilized HFs (*Figure 1—figure supplement*

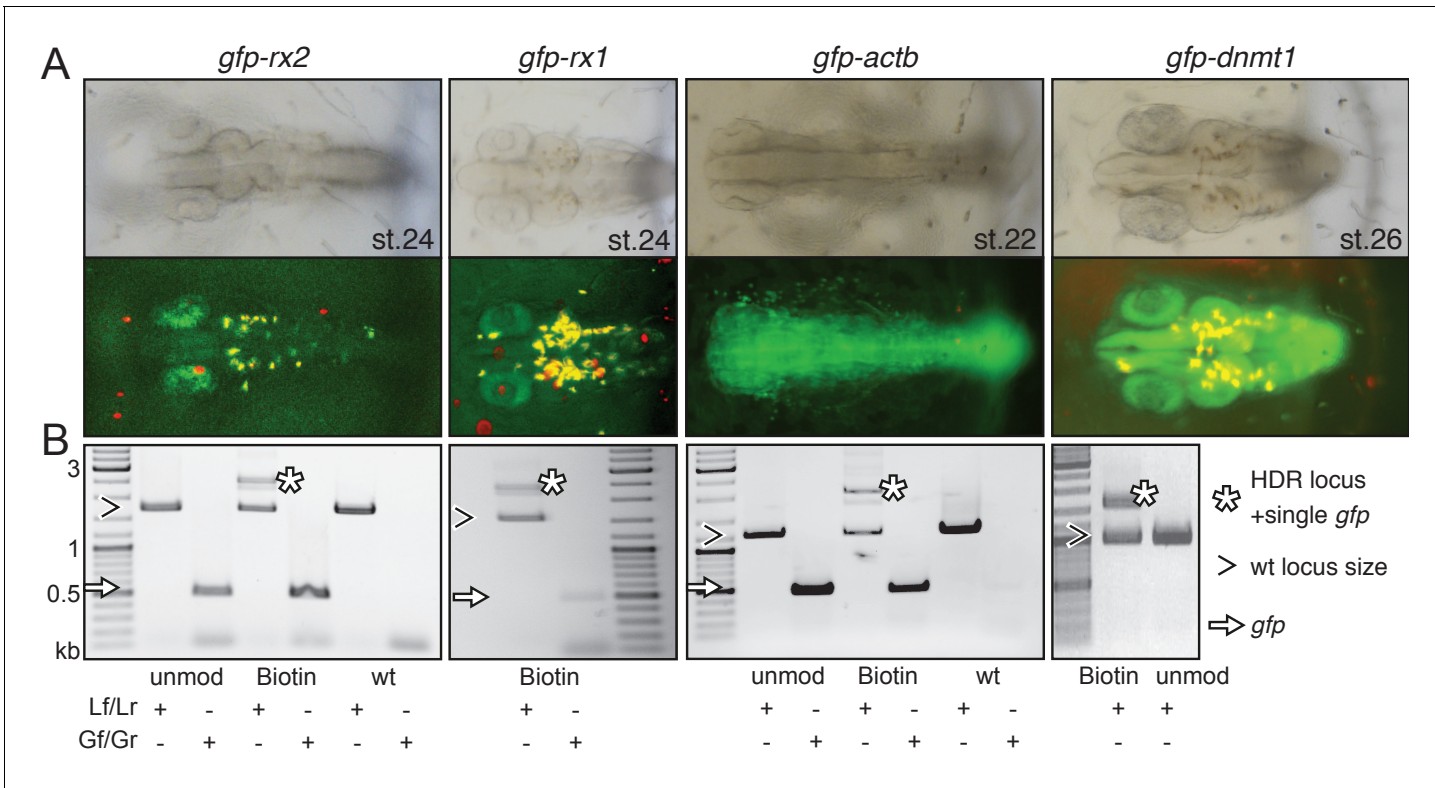

**Figure 2.** Modification of 5' ends of long dsDNA fragments promotes HDR-mediated single-copy integration. (**A**) GFP expression in the respective expression domain after HDR-mediated integration of modified dsDNA *gfp* donor cassettes into *rx2*, *rx1*, *actb* and *dnmt1* ORFs in the injected generation. (**B**) Individual embryo PCR genotyping highlights efficient HDR-mediated single-copy integration of 5'Biotin modified long dsDNA donors, but not unmodified donor cassettes. Locus PCR reveals band size indicative of single-copy *gfp* integration (asterisk) besides alleles without *gfp* integration (open arrowhead). Amplification of *gfp* donor (white arrow) for control.

DOI: https://doi.org/10.7554/eLife.39468.005

The following source data and figure supplements are available for figure 2:

**Figure supplement 1.** Quantification of survival and GFP expression of injected embryos.

DOI: https://doi.org/10.7554/eLife.39468.006

**Figure supplement 1—source data 1.** Quantification of survival and GFP expression of injected embryos.

DOI: https://doi.org/10.7554/eLife.39468.007

**Figure supplement 2.** 5'Biotin modification of long dsDNA donors strongly enhances HDR-mediated integration.

DOI: https://doi.org/10.7554/eLife.39468.008

**Figure supplement 3.** Stable germline transmission of the single-copy HDR-mediated precise *gfp* integration.

DOI: https://doi.org/10.7554/eLife.39468.009

1) and addressed the fusion of the *gfp* donor sequence to the target genes (**Figure 2B**, **Figure 2—figure supplements 2** and **3**).

Employing unmodified donors, the rate of HDR was very low as evidenced by the predominant amplification of the alleles without *gfp* integration (**Figure 2B**, **Figure 2—figure supplement 2**). In strong contrast, the 5' modified long dsDNA donors resulted in efficient HDR already detectable in the injected generation for all targeted loci. For the *gfp* tagging of *rx2*, 6 out of 10 randomly selected, GFP-expressing embryos showed precise HDR-mediated single-copy integration in F0 (**Figure 2—figure supplement 2**) as sequence confirmed by the analysis of the locus (**Figure 2—figure supplement 3**). Thus, 9.5% of injected and surviving zygotes showed precise HDR-mediated single-copy integration (15.8% of the injected zygotes expressed GFP; 60% of those showed the precise single-copy integration; **Supplementary file 1**). For the *gfp* tagging of *actb* 46.5% of the injected zygotes expressed GFP, 35% of which (7 out of 20 randomly selected, GFP-expressing embryos) showed precise HDR-mediated single-copy integration in F0, accounting for 16.3% of the initially injected zygotes (**Supplementary file 1**).

In the case of *dnmt1*, the rate of precise HDR-mediated single-copy integration was even higher, since full gene functionality is required for the progression of development and embryonic survival. Here, strikingly, all GFP-expressing embryo showed the desired perfect integration.

We observed the highest efficiency of HDR targeting by 5' modified dsDNA for all loci tested for 5'Biotin modified donors (**Figure 2—figure supplement 2**). Already in the injected generation we prominently detected and validated the HDR-mediated fusion of the long dsDNA donors with the respective locus (**Figure 2—figure supplement 2**). While still giving rise to a high percentage of HDR events, SpC3 5' modified dsDNA donors also resulted in elevated levels of additional bands indicative for the integration of higher order multimers and NHEJ events (**Figure 2—figure supplement 2**). Even though increasingly popular, in our hands the use of RNPs (Cas9 protein and respective sgRNA) did not even get close to the efficiency achieved by co-injection of Cas9 mRNA and the corresponding sgRNA assessed by gene targeting as described above.

The precise integration detected in the injected generation was successfully transmitted to the next generation (**Figure 3**, **Figure 2—figure supplement 3**, **Figure 3—figure supplement 1**). For *gfp-rx2*, 9% of fish originating from the initially injected embryos successfully transmitted the precisely modified locus to the next generation (15.8% of the injected embryos expressed GFP; 4 out of 7 GFP transmitting founder fish were also transmitting the precise single integration of the *gfp* donor cassette). For *gfp-rx1*, 3.9% of fish originating from the initially injected embryos were transmitting the precise single copy integrate to the next generation (13.5% of injected embryos expressed GFP; 2 out of 7 GFP transmitting founder fish were also transmitting the precise single integration of the *gfp-rx1* donor cassette). For *dnmt1*, the high rate of precise HDR-mediated single-copy integration observed in the injected generation was fully maintained in the transmission to the next generation due to the absolute requirement of a functional/functionally tagged version of the locus. As we found in the course of our study, the precise integration of the *gfp* donor cassette into the *actb* locus results in late embryonic lethality. Consequently, stable transgenic lines could not be established.

To estimate the timepoint of HDR events in the injected embryos, we investigated the actual rate of mosaicism in the germline as reflected by the germline transmission rate. For *gfp-rx2*, this rate ranged from 0.8% up to 12.9% indicating an HDR event earliest at the 4-cell stage (assuming that only a single event occurred per blastomere). In the case of *gfp-rx1*, HDR did not occur at the one-cell stage but rather later (2–8 cell stage as reflected by germline transmission rates of 23.9% and 5.8% respectively). Thus, for both cases, the transmission rates of the perfectly tagged locus reflect a level of mosaicism indicative for an HDR event between the 4- and 32-cell stage.

We addressed the nature of the insertion predicted to be single-copy by a combination of PCR and expression studies. We validated the genomic organization of the *gfp-rx2* (**Figure 3A**) and *gfp-rx1* knock-in in homozygous F2 animals by genomic sequencing (**Figure 2—figure supplement 3C**) and Southern Blot (**Southern, 2006**) analysis (**Figure 3B and B'**, **Figure 3—figure supplement 1**). In both cases, we detected a single band indicative for a single-copy HDR-mediated integration, when probing digested genomic DNA of F2 *gfp-rx2$^{+/+}$* and *gfp-rx1$^{+/+}$* knock-in fish respectively. We used enzyme combinations releasing the *gfp-rx2* region including the HFs (BglII/HindIII, **Figure 3B and B'**) or cutting in the 5' HF (ScaI/HindIII, **Figure 3B,B'**). For *gfp-rx1* we released the 5' flanking region

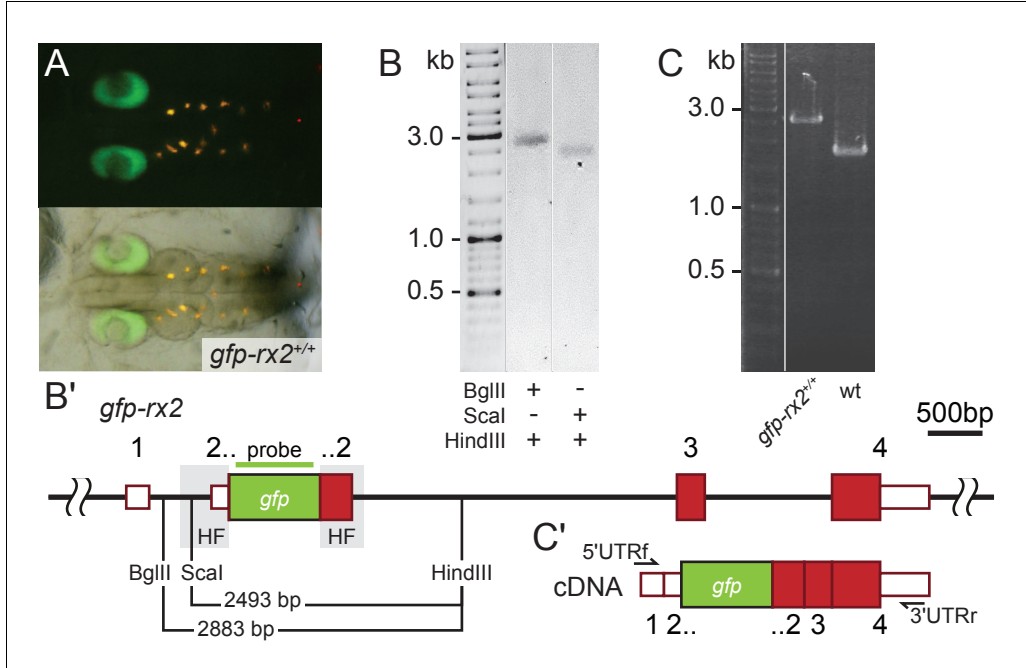

**Figure 3.** Single-copy integration of long dsDNA donor establishes stably transmitted *gfp-rx2* fusion gene. (**A**) F2 homozygous embryos exhibit GFP-Rx2 fusion protein expression in the pattern of the endogenous gene in the retina. (**B**) Southern Blot analysis of F2 *gfp-rx2* embryos reveals a single band for a digestion scheme cutting outside the donor cassette (BglII/HindIII) or within the 5' donor cassette and in intron 2 (ScaI/HindIII) indicating precise single-copy donor integration. (**B'**) Schematic representation of the modified locus indicating the restriction sites and the domain complementary to the probe used in (**B**). (**C**) RT-PCR analysis on mRNA isolated from individual homozygous F3 embryos indicates the transcription of a single *gfp-rx2* fusion mRNA in comparison to the shorter wild-type *rx2* mRNA as schematically represented in (**C'**).

DOI: https://doi.org/10.7554/eLife.39468.010

The following figure supplement is available for figure 3:

**Figure supplement 1.** Stably transmitted single-copy integration of the *gfp-rx1* donor cassette.

DOI: https://doi.org/10.7554/eLife.39468.011

including the donor cassette (HindIII/XmaI) or the respective 3' flanking region and donor cassette (NcoI/EcoRI) (*Figure 3—figure supplement 1*).

This analyses crucially validated the PCR based predictions and sequencing results.

Furthermore, transcript analysis in *gfp-rx2* homozygous F3 embryos exclusively uncovered a single fusion *gfp-rx2* transcript (*Figure 3C and C'*). This molecular analysis confirmed that the 5'Biotin modification of the long dsDNA donor promoted precise single-copy HDR-mediated integration with high efficiency.

Taken together the simple 5'Biotin modification at both ends of long dsDNA donors by conventional PCR amplification presented here provides the means to favor HDR without interfering with the cellular DNA repair machinery.

## Discussion

For efficient recombination the quality of the modified primers, that is the fraction of primers actually labeled with biotin and therefore the quality of the 5' protected long ds DNA PCR product, is essential. The integration of 5' modified PCR fragments larger than 2 kb does in principle not pose a problem. We already successfully integrated cassettes of up to 8.6 kb (data not shown) via HDR by in vivo linearization of the donor plasmid (*Stemmer et al., 2015*). In the approach presented here, the quality of the end-protected PCR product is likely to drop with higher length in part due to the (UV–) nicking in the extraction and purification process. Also, the rapid validation by locus spanning PCR is size limited but eventually Southern blot analysis will resolve the question of a single copy,

perfect integration. Irrespective of the donor cassette size, the PCR cycles used to probe for the successful single integration via HDR must not exceed a total of 30 to avoid misleading in vitro fusion-PCR artifacts (own data and [*Won and Dawid, 2017*]).

As already mentioned earlier, HDR preferentially occurs during the late S/G2 phase of the cell cycle (*Heyer et al., 2010*; *Hustedt and Durocher, 2017*). Varying efficiencies of genome editing via HDR could be due to species-specific differences, in particular of the S/G2 phase of the cell cycle that likely crucially impacts on the rate of HDR. As reported extensively, there is no apparent G2 phase before mid-blastula-transition in amphibia (*Xenopus*) and fish (*Danio rerio*) with the marked exception of the first cleavage, where a G2 phase has been reported (*Kimelman, 2014*). The germ-line transmission rates obtained for the tagging of Rx1 and Rx2 indicated HDR events after the two-cell stage, predominantly at the 4-cell and subsequent stages. The slower cycling of medaka compared to zebrafish results in an extension of the G2 phase in the first cell cycle by more than 150% (total cycle zebrafish: 45 min [*Kimmel et al., 1995*], medaka: 65 min [*Iwamatsu, 2004*]). Assuming a comparable rate of DNA synthesis between the two species, there is more than double the time for S- and M-phase to replicate a genome encompassing roughly a third of the size of zebrafish. This extended cell cycle length is in particular apparent also in the subsequent cycles in medaka (15 min zebrafish [*Kimmel et al., 1995*], 40 min medaka [*Iwamatsu, 2004*]). Interestingly, phosphorylation of RNA polymerase, a prerequisite for the expression of the first zygotic transcripts has been reported to start at the 64cell stage in medaka (*Kraeussling et al., 2011*) and asynchronous divisions, a hallmark of MBT, are observed already at the transition of the 16- to 32-cell stage (*Kraeussling et al., 2011*), suggesting species-specific differences in the onset of zygotic transcription and consequently the lengthening of the cell cycle.

Our approach facilitates the highly efficient detection of HDR events by GFP tagging even in non-essential genes. Other than in non-modified dsDNA donors, the locus-specific expression of GFP is an excellent predictor for precise, single-copy HDR-mediated integration already in the injected generation. This allows an easy selection and results in HDR rates in GFP positive embryos of up to 60% in the injected generation. This rate can even be higher in essential loci, such as *dnmt1*, where the full functionality of the (modified) gene under investigation is required for the survival of the affected cells, tissues, organs or organisms, thus providing the means for effective positive selection for single, fully functional integrates. It is interesting to note that approximately 400 bp flanking the insertion site in 5' and 3' direction are sufficient for HDR. The use of modified universal primers for the two donor template vectors employed allowed a flexible application of the procedure and ensured a high quality of the PCR generated 5' modified long dsDNA donors. Strikingly, the primer-introduced non-homology regions in the very periphery of the long dsDNA donors did not negatively impact on the HDR efficiency.

Taken together, the simplicity and high reproducibility of our proof-of-concept analysis highlight that the presented protection of both 5' ends of long dsDNA donors prevent multimerization and promote precise insertion/replacement of DNA elements, thus facilitating functional studies in basic research as well as therapeutic interventions.

## Materials and methods

**Key resources table**

| Reagent type (species) or resource | Designation | Source or reference | Identifiers | Additional information |
|---|---|---|---|---|
| Strain, strain background (*Oryzias latipes*) | Cab | other | | medaka Southern wild-type population |
| Strain, strain background (*Oryzias latipes*) | *rx2-gfp* | this paper | | |
| Strain, strain background (*Oryzias latipes*) | *rx1-gfp* | this paper | | |

*Continued on next page*

*Continued*

| Reagent type (species) or resource | Designation | Source or reference | Identifiers | Additional information |
|---|---|---|---|---|
| Strain, strain background (*Oryzias latipes*) | *actb-gfp* | this paper | | |
| Strain, strain background (*Oryzias latipes*) | *dnmt1-gfp* | this paper | | |
| Recombinant DNA reagent | *rx2-gfp* donor cassette | this paper | | |
| Recombinant DNA reagent | *rx1-gfp* donor cassette | this paper | | |
| Recombinant DNA reagent | *actb-gfp* donor cassette | this paper | | |
| Recombinant DNA reagent | *dnmt1-gfp* donor cassette | this paper | | |
| Sequence-based reagent | rx2 5'HF f | this paper | | with BamHI restriction site: GCCGGATCCAAGCATGTCAAAACGTAGAAGCG |
| Sequence-based reagent | rx2 5'HF r | this paper | | with KpnI restriction site: GCCGGTACCCATTTGGCTGTGGACTTGCC |
| Sequence-based reagent | rx2 3'HF f | this paper | | with BamHI restriction site: GCCGGATCCCATTTGTCAATGGAC ACGCTTGGGATGGTGGACGAT |
| Sequence-based reagent | rx2 3'HF r | this paper | | with KnpI restriction site: GCCGGTACCTGGACTGGACTGGAAGTTATTT |
| Sequence-based reagent | rx2 sgRNA f | this paper | | substituted nucleotides to facilitate T7 in vitro transcription of the sgRNA oligonucleotides are shown in small letters TAgGCATTTGTCAATGGATACCC |
| Sequence-based reagent | rx2 sgRNA r | this paper | | AAACGGGTATCCATTGACAAATG |
| Sequence-based reagent | rx2 Lf/5'UTRf | this paper | | TGCATGTTCTGGTTGCAACG |
| Sequence-based reagent | rx2 Lr | this paper | | AGGGACCATACCTGACCCTC |
| Sequence-based reagent | actb 5'HF f | this paper | | with BamHI restriction site: GGGGATCCCAGCAACGACTTCGCACAAA |
| Sequence-based reagent | actb 5'HF r | this paper | | with KnpI restriction site: GGGGTACCGGCAATGTCATCATCCATGGC |
| Sequence-based reagent | actb 3'HF f | this paper | | with BamHI restriction site: GGGGATCCGACGACGATATAGCTG CACTGGTTGTTGACAACGGATCTG |
| Sequence-based reagent | actb 3'HF r | this paper | | with KnpI restriction site: GGGGTACCCAGGGGCAATTCTCAGCTCA |
| Sequence-based reagent | actb sgRNA f | this paper | | TAGGATGATGACATTGCCGCAC |
| Sequence-based reagent | actb sgRNA r | this paper | | AAACGTGCGGCAATGTCATCAT |
| Sequence-based reagent | actb Lf | this paper | | GTCCGAGTTGAGGGTGTCTG |
| Sequence-based reagent | actb Lr | this paper | | CATGTGCTCCACTGTGAGGT |
| Sequence-based reagent | dnmt1 5'HF f | this paper | | with SalI restriction site: AATTTGTCGACGCTTTGA CAGTTAACCTACACG |

*Continued on next page*

*Continued*

| Reagent type (species) or resource | Designation | Source or reference | Identifiers | Additional information |
|---|---|---|---|---|
| Sequence-based reagent | dnmt1 5'HF r | this paper | | with AgeI restriction site: AATTTACCGGTCGTAACTGCA AACTAAAAAATAAAAC |
| Sequence-based reagent | dnmt1 3'HF f | this paper | | with SpeI restriction site: AATTTACTAGTATGCCATCCAGAA CGTCCTTATCTCTACCAGACGATG TCAGAAAAAGGTAC |
| Sequence-based reagent | dnmt1 3'HF r | this paper | | with NotI restriction site: AATTTGCGGCCGCCTACACATA TTGTCTGTGATAC |
| Sequence-based reagent | mgfpf | this paper | | with AgeI restriction site: AATTTACCGGTACTAGTACCATG AGTAAAGGAGAAGAACTTTTCAC |
| Sequence-based reagent | mgfpr | this paper | | with SpeI restriction site: AATTTACTAGTCGCGGCTGCACTT CCACCGCCTCCCGATCCGCCACC GCCAGAGCCACCTCCGCCTGAAC CGCCTCCACCGCTCAGGCTAGCTT TGTATAGTTCATCCATGCCATG |
| Sequence-based reagent | dnmt1 sgRNA f | this paper | | substituted nucleotides to facilitate T7 in vitro transcription of the sgRNA oligonucleotides are shown in small letters TAgGACATCGTCTGGCAAAGAC |
| Sequence-based reagent | dnmt1 sgRNA r | this paper | | AAACGTCTTTGCCAGACGATGT |
| Sequence-based reagent | dnmt1 Lf | this paper | | CTCAATGTAAACACTTCGTGTCGCTTC |
| Sequence-based reagent | dnmt1 Lr | this paper | | TTGCATGCATATTCAAAGTTGTCAAAG |
| Sequence-based reagent | rx1 5'HF f | this paper | | with BamHI restriction site: GCCGGATCCGCATCCGAAAGG TAAGGACTGCAAACC |
| Sequence-based reagent | rx1 5'HF r | this paper | | with KpnI restriction site: GCCGGTACCCATGAGAGCG TCTGGGCTCTGACC |
| Sequence-based reagent | rx1 3'HF f | this paper | | with BamHI restriction site: GGCGGATCCCATTTATCAC TCGATACCATGAGCA |
| Sequence-based reagent | rx1 3'HF r | this paper | | with KpnI restriction site: GGCGGTACCTTCCAGTTTA AGAACATCCCCTCT |
| Sequence-based reagent | rx1 sgRNA1 f | this paper | | substituted nucleotides to facilitate T7 in vitro transcription of the sgRNA oligonucleotides are shown in small letters TAggAAATGCATGAGAGCGTCT |
| Sequence-based reagent | rx1 sgRNA1 r | this paper | | AAACAGACGCTCTCATGCATTT |
| Sequence-based reagent | rx1 sgRNA2 f | this paper | | substituted nucleotides to facilitate T7 in vitro transcription of the sgRNA oligonucleotides are shown in small letters TAggCTCTCATGCATTTATCAC |
| Sequence-based reagent | rx1 sgRNA2 r | this paper | | AAACGTGATAAATGCATGAGAG |

*Continued on next page*

*Continued*

| Reagent type (species) or resource | Designation | Source or reference | Identifiers | Additional information |
|---|---|---|---|---|
| Sequence-based reagent | rx1 Lf | this paper | | CTTTGCTGTTTTGAGAATTGCACC |
| Sequence-based reagent | rx1 Lr | this paper | | GAGACCGAACGATGACAATAACAC |
| Sequence-based reagent | pDest f (control) | this paper | | CGAGCGCAGCGAGTCAGTGAG |
| Sequence-based reagent | pDest r (control) | this paper | | CATGTAATACGACTCACTATAG |
| Sequence-based reagent | pDest f mod | this paper | | Asterisks indicate phosphorothioate bonds, '5'moiety' was either 5'Biotin, Amino-dT or Spacer C3. 5'moiety-C*G*A*G*C*GCAGCGAGTCAGTGAG |
| Sequence-based reagent | pDest r mod | this paper | | Asterisks indicate phosphorothioate bonds, '5'moiety' was either 5'Biotin , Amino-dT or Spacer C3. 5'moiety-C*A*T*G*T*AATACGACTCACTATAG |
| Sequence-based reagent | pCS2 f | this paper | | CCATTCAGGCTGCGCAACTG |
| Sequence-based reagent | pCS2 r | this paper | | CACACAGGAAACAGCTATGAC |
| Sequence-based reagent | pCS2 f mod | this paper | | Asterisks indicate phosphorothioate bonds, '5'moiety' was either 5'Biotin, Amino-dT or Spacer C3. 5'moiety-C*C*A*T*T*CAGGCTGCGCAACTG |
| Sequence-based reagent | pCS2 r mod | this paper | | Asterisks indicate phosphorothioate bonds, '5'moiety' was either 5'Biotin, Amino-dT or Spacer C3. 5'moiety-C*A*C*A*C*AGGAAACAGCTATGAC |
| Sequence-based reagent | Gf | this paper | | ATGGCAAGCTGACCCTGAAGTTCATCTGCACCACCGGCAAGC |
| Sequence-based reagent | Gr | this paper | | CTCAGGTAGTGGTTGTCG |
| Sequence-based reagent | gfpf | this paper | | GCTCGACCAGGATGGGCA |
| Sequence-based reagent | gfpr | this paper | | CTGAGCAAAGACCCCAACGAGAAGCGCGATCACATG |
| Sequence-based reagent | gfp probe f | this paper | | GTGAGCAAGGGCGAGGAGCT |
| Sequence-based reagent | gfp probe r | this paper | | CTTGTACAGCTCGTCCATG |

## Fish maintenance

All fish are maintained in closed stocks at Heidelberg University. Medaka (*Oryzias latipes*) husbandry (permit number 35–9185.64/BH Wittbrodt) and experiments (permit number 35–9185.81/G-145/15 Wittbrodt) were performed according to local animal welfare standards (Tierschutzgesetz §11, Abs. 1, Nr. 1) and in accordance with European Union animal welfare guidelines (*Bert et al., 2016*). The fish facility is under the supervision of the local representative of the animal welfare agency. Embryos of medaka of the wild-type Cab strain were used at stages prior to hatching. Medaka was raised and maintained as described previously (*Koster et al., 1997*).

## Donor plasmids

*Rx2* and *actb* template plasmids for *gfp* donor cassette amplification are described in *Stemmer et al. (2015)* and were generated by GoldenGATE assembly into the pGGDestSC-ATG destination vector (addgene #49322) according to *Kirchmaier et al. (2013)*. See *Supplementary file 2* for primers used

to amplify respective homology flanks. The *dnmt1 gfp* plasmid was cloned with homology flanks (5' HF 402 bp, primers dnmt1 5'HF f/dnmt1 5'HF r; 3' HF 405 bp, dnmt1 3'HF f/dnmt1 3'HF r) that were PCR amplified with Q5 polymerase (New England Biolabs, 30 cycles) from wild-type medaka genomic DNA. *mgfp-flexible linker* was amplified with primers mgfpf/mgfpr. The respective restriction enzyme was used to digest the amplicons (5'HF: SalI HF (New England Biolabs), AgeI HF (New England Biolabs); *mgfp-flexible linker*: AgeI HF (New England Biolabs), SpeI HF (New England Biolabs); 3'HF: SpeI HF (New England Biolabs), NotI HF (New England Biolabs)) followed by gel purification (Analytik Jena) and ligation into pCS2+ (*Rupp et al., 1994*) (digested with SalI HF (New England Biolabs), NotI HF (New England Biolabs)). The *rx1 gfp* plasmid was cloned with homology flanks (5'HF 430 bp, primers rx1 5'HF f/rx1 5'HF r; 3' HF 508 bp, rx1 3'HF f/rx1 3'HF r) that were PCR amplified with Q5 polymerase (New England Biolabs, 30 cycles) from wild-type medaka genomic DNA. All primers were obtained from Eurofins Genomics.

## Donor amplification

We designed universal primers that match the pGGDestSC-ATG (*Kirchmaier et al., 2013*) (addgene #49322) or pCS2+ (*Rupp et al., 1994*) backbone encompassing the assembled inserts (i.e. the *gfp* donor cassette). Unmodified control primers (pDest f, pDest r, pCS2 f, pCS2 r) were ordered from Eurofins Genomics. Modified primers obtained from Sigma-Aldrich (pDest f mod, pDest r mod, pCS2 f mod, pCS2 r mod) consist of the same sequences with phosphorothioate bonds in the first five nucleotides and 5'moiety extension: 5'Biotin, 5'Amino-dT or 5'Spacer C3.

The dsDNA donor cassettes were amplified by PCR using 1x Q5 reaction buffer, 200 µM dNTPs, 200 µM primer forward and reverse and 0.6 U/µl Q5 polymerase (New England Biolabs). Conditions used: initial denaturation at 98°C 30 s, followed by 35 cycles of: denaturation at 98°C 10 s, annealing at 62°C 20 s and extension at 72°C 30 s per kb and a final extension step of 2 min at 72°C. The PCR reaction was treated with 20 units of DpnI (New England Biolabs) to remove any plasmid template following gel purified using the QIAquick Gel Extraction Kit (Qiagen, 28706) and elution with 20 µl nuclease-free water.

The *LacZ* cassette of the pGGDestSC-ATG (*Kirchmaier et al., 2013*) (addgene #49322) which served as DIG labelled dsDNA fragment to test in vivo multimerization was amplified via Q5-PCR as above using a mixture of 200 µM dATP, dCTP, dGTP, 170 µM dTTP and 30 µM DIG-dUTP and purified as detailed.

## sgRNA target site selection

*Dnmt1* sgRNAs were designed with CCTop as described in *Stemmer et al. (2015)*. sgRNAs for *rx2* and *actb* were the same as in *Stemmer et al. (2015)*. The following target sites close to the translational start codons were used (PAM in brackets): *rx2* (GCATTTGTCAATGGATACCC[TGG]), *actb* (GGATGATGACATTGCCGCAC[TGG]), *dnmt1* (TGACATCGTCTGGCAAAGAC[AGG]) and *rx1* (AAATGCATGAGAGCGTCT[GGG] and CTCTCATGCATTTATCAC[TGG]). Cloning of sgRNA templates and in vitro transcription was performed as detailed in *Stemmer et al. (2015)*.

## In vitro transcription of mRNA

The pCS2 +Cas9 plasmid was linearized using NotI and the mRNA was transcribed in vitro using the mMessage_mMachine SP6 kit (ThermoFisher Scientific, AM1340).

## Microinjection and screening

Medaka zygotes were injected with 10 ng of DIG-labelled donors and were allowed to develop until 2, 4 and 6 hr post injection. For the CRISPR/Cas9 experiments, medaka zygotes were injected with 5 ng/µl of either unmodified and modified long dsDNA donors together with 150 ng/µl of *Cas9* mRNA and 15–30 ng/µl of the gene-specific sgRNAs. Injected embryos were maintained at 28°C in embryo rearing medium (ERM, 17 mM NaCl, 40 mM KCl, 0.27 mM CaCl$_2$, 0.66 mM MgSO$_4$, 17 mM Hepes). One day post-injection (dpi) embryos were screened for survival, GFP expression was scored at two dpi.

## Southern blot

In order to check for multimerization of unmodified and modified donors, we used a modified Southern Blot approach. In brief, embryos were injected with DIG-labelled donors which were PCR-amplified from pGGDestSC-ATG (addgene #49322) using primers pDest f/pDest r (*LacZ* cassette) harboring either no 5' moiety or one of the following: 5'Biotin, Amino-dT or Spacer C3. 2, 4 and 6 hr post injection, 30 embryos were lysed in TEN buffer plus proteinase K (10 mM Tris pH 8, 1 mM EDTA, 100 mM NaCl, 1 mg/ml proteinase K) at 60°C overnight. DNA was ethanol precipitated after removal of lipids and proteins by phenol-chloroform extraction. Total DNA was resuspended in TE buffer (10 mM Tris HCl pH 8.0, 1 mM EDTA pH 8.0). 200 ng of each sample was run on a 0.8% agarose gel. As a control, 100 pg of uninjected donor PCR product were loaded. The agarose gel was transferred to a nylon membrane overnight using 10x SSC (1.5 M NaCl, 0.15 M $C_6H_5Na_3O_7$) as transfer solution. The cross-linked membrane was directly blocked in 1% w/v blocking reagent (Roche) in 1x DIG1 solution (0.1 M maleic acid, 0.15 M NaCl, pH 7.5) and the labeled DNA was detected using CDP star (Roche) following the manufacturer's instructions.

In order to check for copy number insertions in the *gfp-rx2* and *gfp-rx1* transgenic lines, genomic DNA was isolated as described above from F2 embryos expressing GFP. 10 µg digested genomic DNA were loaded per lane on a 0.8% agarose gel and size fractionated by electrophoresis. The gel was depurinated in 0.25 N HCl for 30 min at room temperature, rinsed with $H_2O$, denatured in 0.5 N NaOH, 1.5 M NaCl solution for 30 min at room temperature and neutralized in 0.5 M Tris HCl, 1.5 M NaCl, pH 7.2 before it was transferred overnight at room temperature onto a Hybond membrane (Amersham). The membrane was washed with 50 mM NaPi for 5 min at room temperature, then crosslinked and pre-hybridized in Church hybridization buffer (0.5 M NaPi, 7% SDS, 1 mM EDTA pH 8.0) at 65°C for at least 30 min. The probe was synthesized from the donor plasmid with primers gfp probe f and gfp probe r using the PCR DIG Probe Synthesis Kit (Roche, 11636090910) and the following PCR protocol: initial denaturation at 95°C for 2 min, 35 cycles of 95°C 30 s, 60°C 30 s, 72°C 40 s and final extension at 72°C 7 min. The probe was boiled in hybridization buffer for 10 min at 95°C and the membrane was hybridized overnight at 65°C. The membrane was washed with preheated (65°C) Church washing buffer (40 mM NaPi, 1% SDS) at 65°C for 10 min, then at room temperature for 10 min and with 1x DIG1% and 0.3% Tween for 5 min at room temperature. The membrane was blocked in 1% w/v blocking reagent (Roche) in 1x DIG1 solution at room temperature for at least 30 min. The membrane was incubated with 1:10,000 anti-digoxigenin-AP Fab fragments (Roche) for 30 min at room temperature in 1% w/v blocking reagent (Roche) in 1x DIG1 solution. Two washing steps with 1x DIG1% and 0.3% Tween were performed for 20 min at room temperature, followed by a 5 min washing step in 1x DIG3 (0.1 M Tris pH 9.5, 0.1 M NaCl) at room temperature. Detection was performed using 6 µl/ml CDP star (Roche).

## Genotyping

Single injected GFP positive embryos were lysed in DNA extraction buffer (0.4 M Tris/HCl pH 8.0, 0.15 M NaCl, 0.1% SDS, 5 mM EDTA pH 8.0, 1 mg/ml proteinase K) at 60°C overnight. Proteinase K was inactivated at 95°C for 10 min and the solution was diluted 1:2 with $H_2O$. Genotyping was performed in 1x Q5 reaction buffer, 200 µM dNTPs, 200 µM primer forward and reverse and 0.012 U/µl Q5 polymerase and 2 µl of diluted DNA sample and the respective locus primers. The conditions were: 98°C 30 s, 30 cycles of 98°C 10 s, annealing for 20 s and 72°C 30 s per kb (extension time used would allow for detecting potential NHEJ events on both ends of the donor) (rx2 Lf/rx2 Lr: 68°C annealing, 90 s extension time; rx1 Lf/rx1 Lr: 66°C annealing, 90 s extension time; actb Lf/actb Lr: 66°C annealing, 84 s extension time; dnmt1 Lf/dnmt1 Lr: 65°C annealing, 90 s extension time) and a final extension of 2 min at 72°C. PCR products were analyzed on a 1% agarose gel.

Diagnostic GFP PCR here: 63°C annealing, 500 bp, 15 s extension

## RT-PCR

Total RNA was isolated from 60 homozygous embryos (stage 32) by lysis in TRIzol (Ambion) and chloroform extraction according to the manufacturer's protocol. RNA was precipitated using isopropanol and resuspended in $H_2O$. cDNA was reverse transcribed with Revert Aid Kit (Thermo Fisher Scientific) after DNAse digestion and inactivation following the manufacturer's instructions. PCR was performed using 5'UTRf, 3'UTRr and Q5 polymerase (New England Biolabs): 98°C 30 s, 35 cycles of

98°C 10 s, annealing 65°C for 20 s and 72°C 210 s and a final extension of 2 min at 72°C. PCR products were analyzed on a 1.5% agarose gel.

## Sequencing

Plasmids and PCR fragments were sequenced with the indicated primers by a commercial service (Eurofins Genomics).

## Acknowledgments

This research was funded through the German Science funding agency (DFG, CRC 873, Project A3 to JW). TTa and ET are members of HBIGS, the Heidelberg Biosciences International Graduate School. We are grateful to M Majewsky, E Leist and A Saraceno for fish husbandry. We thank Oliver Gruss (Bonn) and Daigo Inoue (Yokohama) and all members of the Wittbrodt lab for their critical, constructive feedback on the procedure and the manuscript.

## Additional information

### Funding

| Funder | Grant reference number | Author |
| --- | --- | --- |
| Deutsche Forschungsge-meinschaft | CRC 873,TP A3 | Joachim Wittbrodt |

The funders had no role in study design, data collection and interpretation, or the decision to submit the work for publication.

### Author contributions

Jose Arturo Gutierrez-Triana, Conceptualization, Data curation, Formal analysis, Validation, Investigation, Writing—original draft; Tinatini Tavhelidse, Conceptualization, Data curation, Formal analysis, Supervision, Validation, Investigation, Visualization, Writing—original draft, Writing—review and editing; Thomas Thumberger, Conceptualization, Data curation, Formal analysis, Supervision, Validation, Investigation, Visualization, Methodology, Writing—original draft, Writing—review and editing; Isabelle Thomas, Kerim Anlas, Validation, Investigation; Beate Wittbrodt, Validation, Investigation, Methodology; Tanja Kellner, Methodology; Erika Tsingos, Validation, Methodology; Joachim Wittbrodt, Conceptualization, Formal analysis, Supervision, Funding acquisition, Investigation, Writing—original draft, Project administration, Writing—review and editing

### Author ORCIDs

Tinatini Tavhelidse (iD) http://orcid.org/0000-0002-6103-9019
Thomas Thumberger (iD) https://orcid.org/0000-0001-8485-457X
Erika Tsingos (iD) https://orcid.org/0000-0002-7267-160X
Joachim Wittbrodt (iD) http://orcid.org/0000-0001-8550-7377

### Decision letter and Author response

Decision letter https://doi.org/10.7554/eLife.39468.016
Author response https://doi.org/10.7554/eLife.39468.017

## Additional files

### Supplementary files

• Supplementary file 1. Analysis of injected embryos. Embryos injected with unmodified or modified (5'Biotin, Amino-dT, Spacer C3) long dsDNA *gfp* donor cassettes matching the *rx2*, *actb*, *dnmt1* or *rx1* locus, were scored for GFP expression and survival. Injections without *Cas9* mRNA for control.
DOI: https://doi.org/10.7554/eLife.39468.012

• Supplementary file 2. Oligonucleotides used in this work. Restriction enzyme sites used for cloning of PCR amplicons are indicated in italics. Substituted nucleotides to facilitate T7 in vitro transcription of the sgRNA oligonucleotides are shown in small letters (*Stemmer et al., 2015*). Locus primers forward (Lf) and reverse (Lr) of respective gene loci, *gfp* primers forward (Gf) and reverse (Gr), *gfp* sequencing primers gfpf and gfpr and primers to amplify the *mgfp-flexible linker*, as well as the *gfp* probe for Southern Blot analysis, are given. Asterisks indicate phosphorothioate bonds, '5'moiety' was either 5'Biotin, Amino-dT or Spacer C3 in the pDest f mod, pDest r mod, pCS2 f mod and pCS2 r mod primers.

DOI: https://doi.org/10.7554/eLife.39468.013

• Transparent reporting form
DOI: https://doi.org/10.7554/eLife.39468.014

## Data availability

All data generated or analyzed during this study are included in the manuscript and supporting files. Source data files have been provided for Figure 2-figure supplement 1.

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
