## [Decision Letter]

Thank you for submitting your article "Efficient single-copy HDR by 5' modified long dsDNA donors" for consideration by *eLife*. Your article has been reviewed by 2 peer reviewers, including Alejandro Sánchez Alvarado as the Reviewing Editor and Reviewer #1, and the evaluation has been overseen by Jessica Tyler as the Senior Editor. The following individual involved in review of your submission has agreed to reveal their identity: Manfred Schartl (Reviewer #2).

The reviewers have discussed the reviews with one another and the Reviewing Editor has drafted this decision to help you prepare a revised submission.

Summary:

This well-written manuscript provides a very attractive protocol for efficient gene knock-in with CRISPR/Cas9 through homology-directed repair. The authors used a streamlined strategy (modification of 5' ends of long dsDNA donors with either biotin or carbon spacers) to successfully address a common problem in the field of genome editing: multimerization of donor molecules in vivo and therefore interference with non-homologous end joining (NHEJ). The authors show in particular that 5' modification with biotin leads to a high frequency of single copy HDR events in injected embryos and confirmed that the modified locus was stably transmitted to offspring with remarkably high efficiency for single copy integrations.

If this method is as efficient as reported, precise genome editing by homologous recombination using CRISPR/Cas9 stands to become more the norm than the exception. Therefore, given the likely broad appeal and interest, this method is likely to generate, the following questions must be addressed by the authors:

Essential revisions:

1) In both the Abstract and the main text, authors assert that the new method reported produces "60% efficiency". However, 60% (6 out of 10 GFP positive fish are germline transformed) is the highest efficiency obtained for one of the three genes tested (6 out of 10 GFP positive fish are germline transformed). Unless I missed it, the efficiency for the other two genes was not reported. This information should be provided. If the efficiency is as high as reported, such numbers should be easy to generate and ideally, more genes could have been readily tested.

2) Also, clarification on how the efficiency of transformation was calculated is needed. As it stands, the calculation of efficiency reported in this manuscript is somewhat misleading. A less ambiguous method to report efficiency may be to indicate the minimum number of embryos that need to be injected to get germline transformation. For example, if the injection of 100 embryos gives 60% efficiency, then the efficiency is high. However, if 1000 embryos need to be injected get 60% efficiency, then the efficiency is low.

---

## [Author Response]

1) In both the Abstract and the main text, authors assert that the new method reported produces "60% efficiency". However, 60% (6 out of 10 GFP positive fish are germline transformed) is the highest efficiency obtained for one of the three genes tested (6 out of 10 GFP positive fish are germline transformed). Unless I missed it, the efficiency for the other two genes was not reported. This information should be provided. If the efficiency is as high as reported, such numbers should be easy to generate and ideally, more genes could have been readily tested.

The referees are touching a crucial point of the manuscript that we apparently had not been able to sufficiently highlight in the submitted version. We have taken care to move all the relevant numbers from the supplemental table to the main text and have in addition extended the manuscript and the supplemental table by additional data concerning the results on tagging of Rx1. Related to the number of injected embryos surviving the procedure, efficiencies range from 4-16% .

We have furthermore taken care to fully explain the tagging of *Dnmt1*, an essential gene. Here only unedited embryos or those with a functional fusion protein survive. This results in a much higher tagging efficiency and thus reflects an interesting selection strategy.

2) Also, clarification on how the efficiency of transformation was calculated is needed. As it stands, the calculation of efficiency reported in this manuscript is somewhat misleading. A less ambiguous method to report efficiency may be to indicate the minimum number of embryos that need to be injected to get germline transformation. For example, if the injection of 100 embryos gives 60% efficiency, then the efficiency is high. However, if 1000 embryos need to be injected get 60% efficiency, then the efficiency is low.

Similar to request 1 we now fully explain the rationale behind the efficiency calculation. We relate all efficiencies to the number of surviving, injected embryos (100%). We first screen for injected embryos expressing GFP. Among these we validate the quality integration (in the injected generation) and now relate that number back to the survivors (instead of only relating it back to the number of GFP positive embryos). While those numbers had been presented in the supplemental table of the submitted version, we now consistently calculate all efficiencies relative to the number of injected surviving embryos.

This gives the reader immediate access to the number of embryos to be injected for likely obtaining a perfect HDR mediated gene fusion. In all cases presented, an injection of less than 100 embryos (20 minutes of work) resulted in several founders efficiently transmitting a perfect fusion to the next generation.